# Development of a simple method for differential delivery of volatile anesthetics to the spinal cord of the rabbit

**Peng Zhang, Yao Li, Ting Xu** \* 

Department of Anesthesiology, Sichuan Academy of Medical Sciences & Sichuan Provincial People's Hospital, University of Electronic Science and Technology of China, Chengdu, Sichuan Province, China

\* xt1979@gmail.com

**Data Availability Statement:** All relevant data are within the manuscript and its Supporting Information files.

**Funding:** Peng Zhang (first author) received the Young Talents Foundation of Sichuan Provincial

## Abstract

Emulsified volatile anesthetic can be directly injected into the circulation and eliminated from blood through lungs. Taking advantage of the unique pharmacokinetics of the emulsified volatile anesthetics, we aimed to develop a less traumatic method to differentially deliver them to the spinal cord of rabbit. Sixteen New Zealand White rabbits were randomly assigned to the isoflurane or sevoflurane group. A catheter was placed into the descending aorta, and emulsified isoflurane (8mg/kg/h) or sevoflurane (12mg/kg/h) was given respectively. The concentration and partial pressure of the anesthetics in the jugular and femoral vein were measured. Our results showed that the partial pressure for isoflurane was 3.91 ±1.11 mmHg and 12.61±1.60 mmHg (1.0MAC), and for sevoflurane was 3.89±1.00 mmHg and 19.92±1.84mmHg (1.0MAC), in the jugular vein and femoral vein, respectively. There was significant difference between jugular and femoral vein partial pressure for both isoflurane and sevoflurane groups (both $P < 0.001$). In conclusion, a simple and minimally invasive method has been successfully developed to selectively deliver isoflurane and sevoflurane to the spinal cord in the rabbit. Before the anesthetics taking action on the brain, 69% of isoflurane and 81% of sevoflurane were removed through lungs. This method can be used to investigate sites and mechanisms of volatile anesthetic action.

## Introduction

Volatile anesthetics can induce a variety of reversible, clinically important effects such as amnesia, hypnosis and immobility [1, 2]. To differentiate whether an anesthetic effect comes from brain, spinal cord or both, methods that selectively deliver the anesthetic to target tissues are needed [3, 4]. Previous research has selectively delivered anesthetics to the brain, upper or lower torso of goat, dog and rabbit using cardiopulmonary bypass technology [5–7]. However, our model permits selective perfusion of the spinal cord versus brain that is less costly and labor intensive while obviating most of the trauma associated with prior models.

Rabbits have a unique spinal cord circulation in that each spinal cord segment is supplied by a single corresponding radicular artery arising from the aorta [8, 9]. In addition, the blood

People's Hospital (2017QN10). The project was supported by this foundation.

**Competing interests:** The authors have declared that no competing interests exist.

supply to the thoracolumbar region (below $T_3$) of spinal cord originates from the thoracic and abdominal aorta in the rabbit [10–12]. Volatile anesthetics can be dissolved in an emulsifiying agent, which produces anesthesia as effectively as a vaporized and inhaled one [13, 14]. Similar to intravenous drugs, emulsified volatile anesthetics can be directly injected into the circulation, but eliminated through lungs [15, 16]. In this report, we aimed to establish a simpler, less labor intensive, and less traumatic method that can be used to selectively deliver volatile anesthetics (isoflurane and sevoflurane) to the spinal cord in rabbit.

## Materials and methods

### Emulsified isoflurane and sevoflurane preparation

Isoflurane and sevoflurane emulsion were prepared according to a previously described formula [17, 18]. In brief, 1.6mL of liquid isoflurane or sevoflurane (Abbott, Shanghai, China) and 18.4mL of 30% intralipid (Baxter, Suzhou, China) were injected into a sealed sterile vial. Then the vial was violently shaken for 10min by a vibrator to solubilize isoflurane or sevoflurane into the intralipid. The 8% emulsified isoflurane and sevoflurane were stored in 4˚C refrigerator before use.

### Animal preparation and surgical procedures

The study protocol was approved by the Institutional Animal Care and Use Committee of Sichuan Provincial People's Hospital. Sixteen New Zealand White rabbits (male and female) weighing 2.0–3.0kg were randomly assigned to the isoflurane or sevoflurane group (8 each). After intravenous injection of 30 mg/kg pentobarbital sodium into the left marginal ear vein, an ID 3.0# cuffed endotracheal tube was inserted. Before intubation, 1% dicaine gel was coated to the endotracheal tube for airway topical anesthesia, which could increase the tolerance for endotracheal tube [18, 19]. All rabbits were mechanically ventilated with 95% $O_2$ using an animal ventilator (Chengdu Techman Software Co.Ltd, Chengdu, China), with the tidal volume 8ml/kg, 40 breaths/min, to maintain the arterial carbon dioxide pressure (PaCO2) between 35 to 45 mmHg. Local anesthetic (1% lidocaine+0.5% ropivacaine) were injected into the skin before surgical exposure. A 22-gauge IV catheter (BD company, Sandy Utach, USA) was individually inserted into the left central ear and femoral arteries respectively to monitor the blood pressure in the upper and lower torso of the rabbit.

An epidural catheter (TuoRen Medical Instrument Co., Ltd, Xinxiang, China) was placed into the descending aorta from the right femoral artery (Fig 1). It was used to deliver emulsified isoflurane or sevoflurane to the spinal cord. The change of resistance was monitored during the catheter passing cranially along the aorta. The catheter would be withdrawn 0.5–1.0 cm, when the increased resistance was recorded. The placement of catheter tip was confirmed by autopsy after the experiment. In our study, the rabbit would be excluded from data analysis if the tip was at the aortic arch level. The length of catheter between catheter tip and inguinal fold was measured.

The infusion rate was 8mg/kg/h in the isoflurane group and 12mg/kg/h in the sevoflurane group by a microinfusion pump. When the end-tidal isoflurane concentration ($E_T$ISO) or end-tidal sevoflurane concentration ($E_T$SEVO) was stable and maintained for 20min, emulsified volatile anesthetic was considered to reach steady state condition [15]. There were two methods to avoid re-breaching of isoflurane or sevoflurane into the brain: (a) a higher inspiratory flow 2L/min (normal minute ventilation (MV) of rabbit is about 1L/min) (b) an anesthetic absorber made of activated charcoal in the inspiratory limb of the circuit.

Lactated Ringer's solution (6ml/kg/h) was administered through the left marginal ear vein in all rabbits. Rectal temperature was monitored and maintained at 37.0±1.0˚C. During the

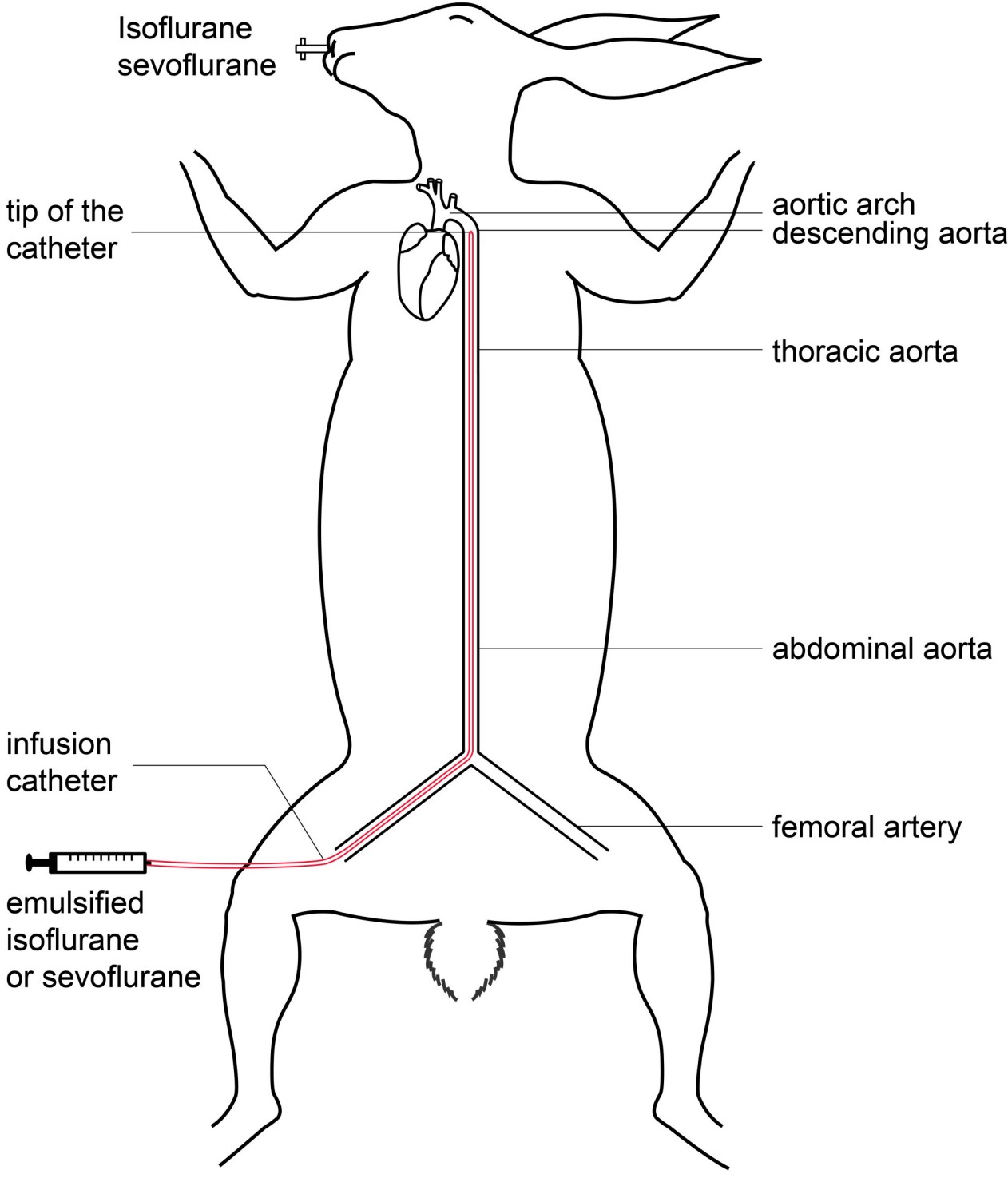

**Fig 1. Diagram of the differential anesthetic delivery rabbit model.**

experiment, heart rate (HR), mean arterial pressure (MAP), electrocardiograph (ECG) and pulse oxygen saturation (SpO$_2$) were measured with the BL-420E+ Data Acquisition and Analysis System (Chengdu Techman Software Co.Ltd, Chengdu, China). The E$_T$ISO, E$_T$SEVO and

$E_T CO_2$ (end-tidal $CO_2$) were monitored with an anesthetic gas analyzer M1026B (Philips Medizin Systeme Boblingen GmbH, Boblingen, Germany). Isoflurane or sevoflurane 1.0 MAC was permitted during the experiment, and was discontinued to wash out volatile anesthetics, 15-20min before the infusion of the emulsified volatile anesthetics.

### Collection of blood samples and gas chromatograph

Once the end-tidal concentration of isoflurane or sevoflurane was stable and maintained for 20min, 5-6ml blood was rapidly withdrawn from the jugular and femoral vein to determine the isoflurane or sevoflurane blood concentration (C). Samples were analyzed using a gas chromatograph Agilent 4890D (Tegent Technology Ltd., Hong Kong, China) with two-stage headspace equilibration methods [15]. The partial pressure of isoflurane and sevoflurane (P) in different blood samples were calculated by the equation of $P = (C/\lambda_{b/g} \times 760$ mmHg) ($\lambda_{b/g} =$ blood/gas partition coefficient). The C and P of all samples were measured by a technician who was blinded to the samples. In our previous study, the partial pressure of 1.0 MAC (minimum alveolar concentration) isoflurane was $11.66 \pm 1.10$ mmHg and the partial pressure of 1.0 MAC sevoflurane was $18.86 \pm 1.12$ mmHg.

### Statistical analysis

Statistical analyses were performed using SPSS 22.0. Each continuous variable was analyzed for its normal distribution and reported as mean ± SD. Repeated measures ANOVA was used to evaluate MAP and HR among different time points in two groups. Other continuous variable data (the weight of the rabbit, the length of the catheter and the partial pressure of isoflurane and sevoflurane) were analyzed by student's t test. $\chi^2$ test was used to evaluate the position of catheter tips. Pearson correlation coefficient was used to measure the linear relationship between $\lambda_{b/g}$ and the volume (mL) of the emulsified isoflurane or sevoflurane consumed. A $P$ value of less than 0.05 was considered statistically significant. A Bonferroni correction was made when necessary to correct for multiple testing.

### Results

Full data sets were collected from sixteen rabbits. The mean weight of the rabbits was $2.38 \pm 0.26$ kg in the isoflurane group and $2.46 \pm 0.24$ kg in the sevoflurane group, which were not significantly different ($P = 0.532$). The mean lengths of the inserted catheter (between catheter tip and inguinal fold) were $25.13 \pm 1.07$ cm in the isoflurane group and $25.36 \pm 0.86$ cm in the sevoflurane group. In the isoflurane group, two catheters (25%) were located at $T_2$ and six (75%) at $T_3$. In the sevoflurane group, four catheters (50%) were placed at $T_2$ and four (50%) at $T_3$ (Table 1). There was no significant difference with regard to the length of the catheter ($P = 0.608$) and the position of the catheter tips ($P = 0.631$) between two groups.

No significant difference was found regarding MAP of the central ear artery among three different time points in two groups ($P = 0.169$ for isoflurane group; $P = 0.390$ for sevoflurane group). But MAP of the femoral artery at the midpoint of the infusion was significantly

**Table 1. The position of catheter tips in two groups.**

| Group | $T_2$ | $T_3$ | Total |
|---|---|---|---|
| Isoflurane | 2 (25%) | 6 (75%) | 8 |
| Sevoflurane | 4 (50%) | 4 (50%) | 8 |
| Total | 6 | 10 | 16 |
| $P$ value | 0.608 | | |

**Table 2. The mean atrial pressure (MAP, mmHg) of femoral and central ear artery among different time points in two groups (mean ± SD).**

| Group | Position | After induction | Midpoint of infusion | End of the study |
|---|---|---|---|---|
| Isoflurane | Central ear artery | 90.4±2.3 | 86.3±5.5 | 87.5±4.5 |
| | Femoral artery | 92.3±3.6 | 74.6±6.1[#&] | 88.4±4.4 |
| Sevoflurane | Central ear artery | 88.1±5.1 | 84.6±7.5 | 84.3±5.4 |
| | Femoral artery | 90.3±5.0 | 71.4±4.3[#&] | 86.1±4.8 |

[#]$P<0.001$, midpoint of infusion vs after induction

[&]$P<0.001$, midpoint of infusion vs end of the study

reduced compared with the other two time points (after induction and at the end of the study) in two groups (both $P<0.001$). The trend of MAP change between central ear artery and femoral artery was significantly different during the whole experiment in both groups ($P<0.001$). However, there was no significant difference in the MAP between central ear artery and femoral artery during the whole experiment in two groups ($P = 0.062$ for isoflurane group; $P = 0.169$ for sevoflurane group) (Table 2). No significant difference was found with regard to HR among three time points in the isoflurane group ($P = 0.091$) and sevoflurane group ($P = 0.076$) (Table 3).

The partial pressure of isoflurane was 3.91±1.11 mmHg in the jugular vein and 12.61±1.60 mmHg (1.0MAC) in the femoral vein. The partial pressure of sevoflurane was 3.89±1.00 mmHg in the jugular vein and 19.92±1.84mmHg (1.0MAC) in the femoral vein. The partial pressure in the jugular and femoral vein was significantly different for both the isoflurane and sevoflurane groups (both $P<0.001$). The ratio of partial pressure of isoflurane and sevoflurane between jugular vein and femoral vein is shown in the Table 4.

There was significant positive correlation between $\lambda_{b/g}$ and volume of the emulsified anesthetics delivered (r = 0.935, $P<0.001$ for isoflurane group; r = 0.919, $P = 0.001$ for sevoflurane group). The Linear regression equation was y = 0.0869x+0.9688 ($R^2 = 0.8723$) in the isoflurane group (Fig 2) and y = 0.0526x+0.3889 ($R^2 = 0.8452$) in the sevoflurane group (Fig 3) (y = volume and x = $\lambda_{b/g}$).

## Discussion

Rabbit, with its medium size and homosegmental blood supply to the spinal cord, is an ideal animal for selective spinal cord anesthetic delivery [10–12]. With the catheter inserted into the descending aorta, isoflurane and sevoflurane emulsion were able to be preferentially delivered to the thoracolumbar (below $T_3$) region of the spinal cord. Before acting on the brain, the majority of anesthetic was eliminated through lungs. Using this model, 69% of isoflurane, or 81% of sevoflurane was eliminated by the lungs before reaching the brain. Therefore, the torso circulation below the $T_3$ level was selectively perfused with anesthetic compared to the upper spinal cord and brain in our study.

When volatile anesthetic reached a steady state condition in the body, the partial pressure in the alveolar, blood and central nervous system is presumed to be in equilibrium with one another [15, 20]. In our study, to be specific, the partial pressure in the jugular and femoral

**Table 3. The heart rate (HR, beats/min) among different time points in two groups (mean ± SD).**

| Group | After induction | Midpoint of infusion | End of the study |
|---|---|---|---|
| Isoflurane | 203±11 | 201±5 | 212±11 |
| Sevoflurane | 215±11 | 207±9 | 220±13 |

**Table 4. The partial pressure of isoflurane and sevoflurane (mmHg) in the jugular (P_j) and femoral vein (P_f) in two groups (mean ± SD).**

| Group | $P_j$ jugular vein | $P_f$ femoral vein | Ratio1 ($P_j$/ $P_f$)% | Ratio2 ($P_f$/ $P_j$) | P value |
|---|---|---|---|---|---|
| Emulsified isoflurane (8mg/kg/h) | 3.91±1.11 | 12.61±1.60 | 31.28±9.17% | 3.48±1.13 | <0.001 |
| Emulsified sevoflurane (12mg/kg/h) | 3.89±1.00 | 19.92±1.84 | 19.40±4.46% | 5.34±1.18 | <0.001 |

veins reflect that in the brain and spinal cord, respectively, under a steady state condition. Our results showed that the isoflurane partial pressure in the femoral vein was about 3.5-fold of that in the jugular vein, and the ratio of sevoflurane was 5.3-fold. In other words, the partial pressure of isoflurane and sevoflurane in the spinal cord were 3.5-fold and 5.3-fold of that in the brain tissue respectively. Thus, our model could differentially deliver emulsified isoflurane and sevoflurane to the spinal cord in rabbit.

In *Yang*'s goat model [18], using similar technologies, emulsified isoflurane was selectively delivered to the spinal cord, with an approximate 46% reduction in the isoflurane partial pressure in the brain [18]. In our present study, brain concentrations of isoflurane and sevoflurane were only 31% and 19% of the spinal cord concentrations, respectively. Therefore, our rabbit model was more selective compared to the previous goat model using emulsified isoflurane. There were several possible reasons: (a) The oxygen consumption per kilogram of rabbit is larger than that of goat, and MV/kg of rabbit is about 2 times than that of goat [21]. It has established that increasing MV might contribute to remove volatile anesthetics from blood via lungs [22]; (b)Two methods mentioned (higher inspiratory flow and anesthetic absorber) were

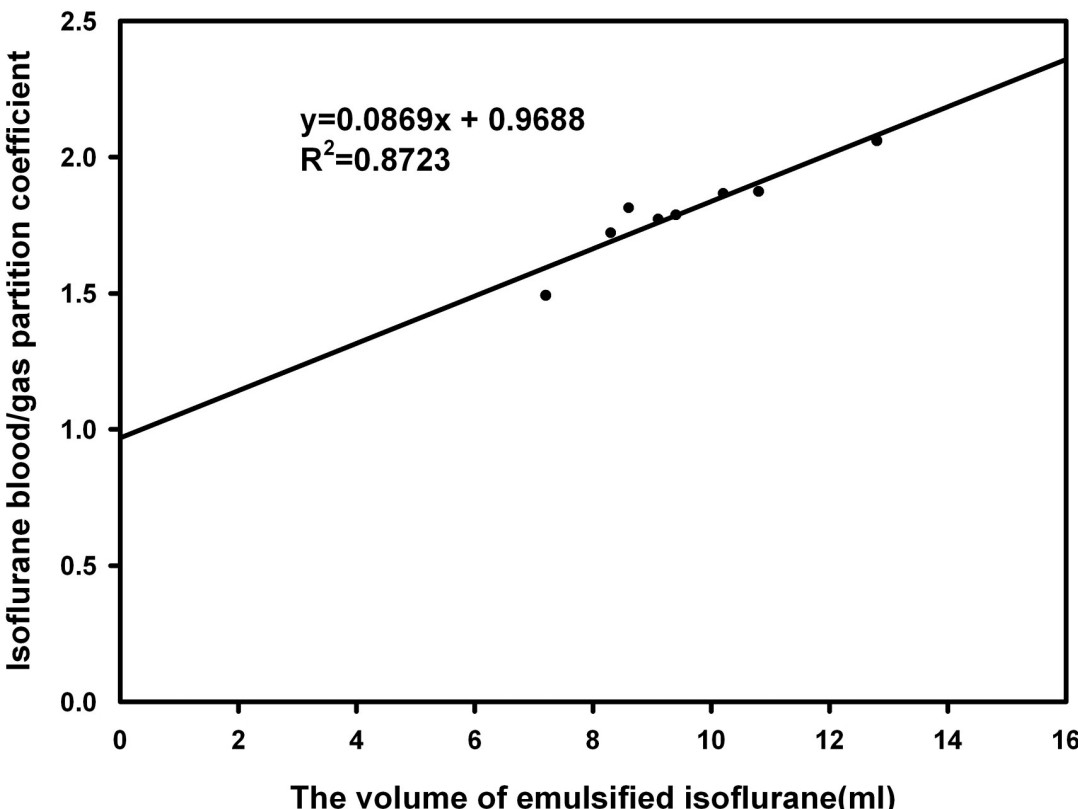

**Fig 2. The positive correlation between isoflurane blood/gas partition coefficient and the volume of emulsified isoflurane delivered.**

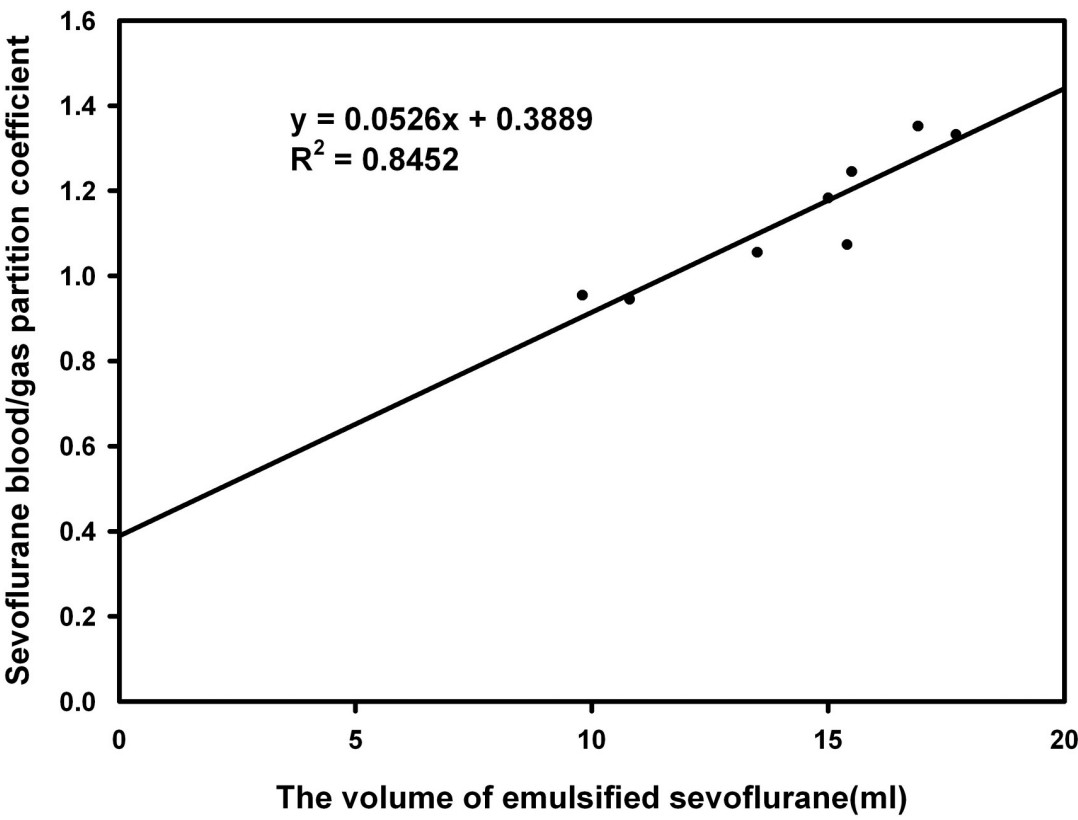

**Fig 3. The positive correlation between sevoflurane blood/gas partition coefficient and the volume of emulsified sevoflurane delivered.**

used to avoid re-breaching of isoflurane or sevoflurane; (c) In our study, sevoflurane was more rapidly eliminated from body than isoflurane, because of its lower blood gas solubility ($\lambda_{b/g}$ of sevoflurane = 0.069 versus $\lambda_{b/g}$ of isoflurane = 1.37) [23].

It is well known that the $\lambda_{b/g}$ of 30% intralipid is remarkably larger than the $\lambda_{b/g}$ of both isoflurane and sevoflurane. So $\lambda_{b/g}$ of isoflurane and sevoflurane would significantly increase with infusion of lipid emulsion in our study. Our results indicated that infusion of 1 ml emulsified volatile anesthetics increased $\lambda_{b/g}$ by 0.0869 for isoflurane and by 0.0526 for sevoflurane at 37˚C. Some discrepancies in the literature showed that 1 ml of 8% emulsified isoflurane raised $\lambda_{b/g}$ by 0.0176 in the goat [18] and 0.0139 in the dog [15]. Obviously, the blood volume of rabbit is significantly less than that of goat or dog [22]. Based on "volume fraction partition coefficient" theory, when 1ml of lipid emulsion was administrated to those three animals, the increase of rabbit $\lambda_{b/g}$ of isoflurane might be larger than that of goat or dog [24]. However, there was no report published regarding $\lambda_{b/g}$ of sevoflurane.

The limitations of our study are as follows: the volatile anesthetics (VAs) concentration in the brain can not be regulated, and the lower VAs partial pressure in the brain (0.2–0.3MAC) could keep the rabbits awake during the infusion period, which might bring animal welfare concerns. However, we believe there are three reasons to eliminate most of the animal welfare issues. (i) In our model, all rabbits were under isoflurane or sevoflurane general anesthesia administrated via a vaporizer for most of the procedure, except the wash-out and infusion periods. Because VA concentration in the brain was equivalent to 0.2–0.3MAC, the rabbits were considered to be hypnotized or sedated during the infusion period [22]. (ii) Given the

minimally invasive technique used, airway topical anesthesia and wound local anaesthetics infiltration can provide extra analgesia and enhance the tolerance to pain stimulus [19, 25, 26]. (iii) Rabbits have a natural tolerance for restraint and can hence remain immobile in a confined space [27]. Considering this unique behavior, it is fair to assume that even a partially conscious rabbit can uneventfully undergo a cerebral MRI and abdominal ultrasound [27, 28]. In our study, during the wash-out and infusion periods, instead of being fixed on the bench, the animal was unlocked and wrapped in a surgical towel with slight manual restrain, which not only helped to relieve pain and alleviate stress, but also made the animal immobile. All rabbits stayed immobile and did not display any signs of distress or discomfort during the procedure.

In our study, because of the limited blood supply to the spinal cord, only the thoracolumbar region (below $T_3$), not the entire spinal cord, could be selectively anesthetized during aortic delivery of emulsified isoflurane and sevoflurane. However, our method still has several advantages. Firstly, because of less trauma, the integrity of cerebral and spinal cord circulation remains intact in this rabbit model [6, 7, 29]. Secondly, unlike bypass models that need special and expensive equipment, this rabbit model is simple and less expensive. At the end of the study, all rabbits were awake and no fatalities occurred. This confirmed that infusion of emulsified isoflurane or sevoflurane produced a safe and reversible anesthesia.

In conclusion, a simple method has been successfully established that permits differential delivery of isoflurane and sevoflurane to the spinal cord. This model was found to eliminate 69% isoflurane and 81% sevoflurane from blood via lungs, before acting on the brain. Taking advantage of the unique homosegmental blood supply to the spinal cord in rabbit, this model can also be used to preferentially deliver anesthetics to the thoracic, lumbar or sacro-coccygeal spinal cord, thereby permitting investigation of the effects of anesthetics on different spinal segments.

## Supporting information

**S1 Data. Relevant data underlying the findings described in manuscript.**
(XLS)

## Author Contributions

**Data curation:** Peng Zhang.

**Funding acquisition:** Peng Zhang.

**Methodology:** Peng Zhang, Yao Li.

**Writing – original draft:** Peng Zhang.

**Writing – review & editing:** Ting Xu.

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
