## [Decision Letter · Decision Letter 0]

20 Nov 2019

PONE-D-19-27037

Development of a simple method for differential delivery of volatile anaesthetics to the spinal cord of the rabbit

PLOS ONE

Dear Peng Zhang, Yao Li and Ting Xu,

Thank you for submitting your manuscript to PLOS ONE. After careful consideration, we feel that it has merit but does not fully meet PLOS ONE’s publication criteria as it currently stands. Therefore, we invite you to submit a revised version of the manuscript that addresses the points raised during the review process.

I would like you to pay particular attention to the reviewer's comment regarding animal welfare concerns. It is very important that the authors discuss this concern both in the rebuttal letter and in the revised manuscript.

We would appreciate receiving your revised manuscript by January 2 2020. To enhance the reproducibility of your results, we recommend that if applicable you deposit your laboratory protocols in protocols.io, where a protocol can be assigned its own identifier (DOI) such that it can be cited independently in the future. For instructions see: http://journals.plos.org/plosone/s/submission-guidelines#loc-laboratory-protocols

We look forward to receiving your revised manuscript.

Kind regards,

Anne Lee Solevåg, M.D., Ph.D.

Academic Editor

PLOS ONE

Journal Requirements:

2. To comply with PLOS ONE submissions requirements, please provide methods of sacrifice in the Methods section of your manuscript.

Reviewers' comments:

Reviewer's Responses to Questions

**Comments to the Author**

1. Is the manuscript technically sound, and do the data support the conclusions?

Reviewer #1: Yes

2. Has the statistical analysis been performed appropriately and rigorously? 

Reviewer #1: Yes

3. Have the authors made all data underlying the findings in their manuscript fully available?

Reviewer #1: No

4. Is the manuscript presented in an intelligible fashion and written in standard English?

Reviewer #1: No

5. Review Comments to the Author

Reviewer #1: The methods manuscript by Zhang et al uses a rabbit mode to differentially deliver volatile anesthetics to the brain versus spinal cord using emulsified agents infused into the descending aorta. Although this has been achieved in prior studies using brain bypass, the main advantage of this model is that it is simple and minimally invasive. However the authors statements that prior models suffered from "CNS damage" is not supported at least in prior goat models where MAC values remained normal before and following bypass, or when bypass was implemented and equal amounts of anesthetic were delivered to the brain and spinal cord. Therefore these comments should be omitted or toned down, and rather the authors should highlight the real advantage of being simpler, less labor intensive and less costly.

The reported differences in partial pressures of volatile anesthetics in the femoral vein ("spinal cord") were 3.5 to 5 fold greater than in jugular vein ("brain") which is very good separation similar to bypass models. However, this model is limited in that unlike bypass models, anesthetic cannot be regulated to the brain. THIS PRESENTS A SERIOUS ANIMAL WELFARE CONCERN, especially for VA with low solubility, like sevoflurane used in the present study. That is, with peri-MAC concentrations delivered to the spinal cord, anesthetic concentrations to the brain reach low enough concentrations to permit the rabbits to become conscious during the experiment. This welfare concern and the fact that brain anesthetic cannot be regulated is a major limitation to the model the authors failed to mention.

The minimal invasive techniques makes this less concerning, however it is still concerning. Perhaps if the rabbits were acclimated and trained to lie quietly in a restrainer with the mask on before conducting the procedure, this would eliminate most of the animal welfare issues. However I cannot say if this is within a rabbits behavioural repertoire.

Last, while I am sympathetic to the language barriers non-english speaking groups face, the manuscript should be reviewed and edited by a fluent english writer BEFORE submission. While some parts are written fairly well, I still extensively edited the manuscript (see attached file) for basic syntax, grammar and clarity. It is not appropriate to place this burden on a scientific reviewer, and moreover the authors should have a vested interest in submitting a comprehensible manuscript to maximize chances of being accepted.

6. PLOS authors have the option to publish the peer review history of their article (what does this mean?). If published, this will include your full peer review and any attached files.

Reviewer #1: No

---

## [Decision Letter · Decision Letter 1]

22 Jan 2020

PONE-D-19-27037R1

Development of a simple method for differential delivery of volatile anesthetics to the spinal cord of the rabbit

PLOS ONE

Dear Dr. Xu Ting Xu,

Thank you for submitting your manuscript to PLOS ONE. After careful consideration, we feel that it has merit but does not fully meet PLOS ONE’s publication criteria as it currently stands. Therefore, we invite you to submit a revised version of the manuscript that addresses the points raised during the review process.

ACADEMIC EDITOR:

Reviewer 1 asks for minor edits. The manuscript will be accepted for publication in PLOS ONE provided that the authors can make these changes:

Overall the authors have addressed comments and concerns, with some minor ones remaining although I feel these do not require re-review:

The manuscript still needs some minor English language editing including the revised text, however it is much improved.

The authors make good points about animal welfare concerns however the comments are mostly speculative without reference to what they actually observed at the relevant timepoints. Please add something to the effect "animals did not display signs of distress or discomfort during the infusion and post-infusion time periods" (assuming this was true).

We would appreciate receiving your revised manuscript by January 31 2020. To enhance the reproducibility of your results, we recommend that if applicable you deposit your laboratory protocols in protocols.io, where a protocol can be assigned its own identifier (DOI) such that it can be cited independently in the future. For instructions see: http://journals.plos.org/plosone/s/submission-guidelines#loc-laboratory-protocols

We look forward to receiving your revised manuscript.

Kind regards,

Anne Lee Solevåg, M.D., Ph.D.

Academic Editor

PLOS ONE

Additional Editor Comments (if provided):

Reviewer 1 asks for minor edits. The manuscript will be accepted for publication in PLOS ONE provided that the authors can make these changes:

Overall the authors have addressed comments and concerns, with some minor ones remaining although I feel these do not require re-review:

The manuscript still needs some minor English language editing including the revised text, however it is much improved.

The authors make good points about animal welfare concerns however the comments are mostly speculative without reference to what they actually observed at the relevant timepoints. Please add something to the effect "animals did not display signs of distress or discomfort during the infusion and post-infusion time periods" (assuming this was true).

Reviewers' comments:

Reviewer's Responses to Questions

**Comments to the Author**

1. If the authors have adequately addressed your comments raised in a previous round of review and you feel that this manuscript is now acceptable for publication, you may indicate that here to bypass the “Comments to the Author” section, enter your conflict of interest statement in the “Confidential to Editor” section, and submit your "Accept" recommendation.

Reviewer #1: (No Response)

2. Is the manuscript technically sound, and do the data support the conclusions?

Reviewer #1: Yes

3. Has the statistical analysis been performed appropriately and rigorously? 

Reviewer #1: Yes

4. Have the authors made all data underlying the findings in their manuscript fully available?

Reviewer #1: Yes

5. Is the manuscript presented in an intelligible fashion and written in standard English?

Reviewer #1: Yes

6. Review Comments to the Author

Reviewer #1: Overall the authors have addressed comments and concerns, with some minor ones remaining although I feel these do not require re-review:

The manuscript still needs some minor English language editing including the revised text, however it is much improved.

The authors make good points about animal welfare concerns however the comments are mostly speculative without reference to what they actually observed at the relevant timepoints. Please add something to the effect "animals did not display signs of distress or discomfort during the infusion and post-infusion time periods" (assuming this was true).

7. PLOS authors have the option to publish the peer review history of their article (what does this mean?). If published, this will include your full peer review and any attached files.

Reviewer #1: No

---

## [Author Response · Author response to Decision Letter 1]

30 Jan 2020

We agree with the reviewer and have performed some modifications in our revised manuscript. Three references (reference 25, 26, 28) was added in the Discussion section to support our results. In addition, the sentence “All rabbits stayed immobile and did not display any signs of distress or discomfort during the procedure” was also added (lines 225-227). Furthermore, we have had our revised manuscript edited by a native English-speaking editor to improve the language quality.

---

## [Editor Report · Decision Letter 2]

3 Feb 2020

Development of a simple method for differential delivery of volatile anesthetics to the spinal cord of the rabbit

PONE-D-19-27037R2

Dear Dr. Ting Xu,

We are pleased to inform you that your manuscript has been judged scientifically suitable for publication and will be formally accepted for publication once it complies with all outstanding technical requirements.

With kind regards,

Anne Lee Solevåg, M.D., Ph.D.

Academic Editor

PLOS ONE
---

## [Editor Report · Acceptance letter]

10 Feb 2020

PONE-D-19-27037R2 

Development of a simple method for differential delivery of volatile anesthetics to the spinal cord of the rabbit 

Dear Dr. Xu:

I am pleased to inform you that your manuscript has been deemed suitable for publication in PLOS ONE. Congratulations! Your manuscript is now with our production department. 

With kind regards,

on behalf of

Dr. Anne Lee Solevåg 

Academic Editor

PLOS ONE